# Global change differentially modulates Caribbean coral physiology

Colleen B. Bove [1,2]*, Sarah W. Davies[2], Justin B. Ries[3], James Umbanhowar[1,4], Bailey C. Thomasson[4,5], Elizabeth B. Farquhar[1,6], Jess A. McCoppin[4], Karl D. Castillo[1,7]

**1** Environment, Ecology, and Energy Program, The University of North Carolina at Chapel Hill, Chapel Hill, North Carolina, United States of America, **2** The Department of Biology, Boston University, Boston, Massachusetts, United States of America, **3** Department of Marine and Environmental Sciences, Northeastern University, Nahant, MA, United States of America, **4** The Department of Biology, The University of North Carolina at Chapel Hill, Chapel Hill, North Carolina, United States of America, **5** Coral Restoration Foundation, Key Largo, Florida, United States of America, **6** Center for Marine Science, University of North Carolina Wilmington, Wilmington, NC, United States of America, **7** Department of Earth, Marine and Environmental Sciences, The University of North Carolina at Chapel Hill, Chapel Hill, North Carolina, United States of America

\* colleenbove@gmail.com

**Data Availability Statement:** All data and code used in this manuscript are archived at Zenodo (10.5281/zenodo.5093907) and can be freely accessed on GitHub (github.com/seabove7/Bove_

## Abstract

Global change driven by anthropogenic carbon emissions is altering ecosystems at unprecedented rates, especially coral reefs, whose symbiosis with algal symbionts is particularly vulnerable to increasing ocean temperatures and altered carbonate chemistry. Here, we assess the physiological responses of three Caribbean coral (animal host + algal symbiont) species from an inshore and offshore reef environment after exposure to simulated ocean warming (28, 31˚C), acidification (300–3290 µatm), and the combination of stressors for 93 days. We used multidimensional analyses to assess how a variety of coral physiological parameters respond to ocean acidification and warming. Our results demonstrate reductions in coral health in *Siderastrea siderea* and *Porites astreoides* in response to projected ocean acidification, while future warming elicited severe declines in *Pseudodiploria strigosa*. Offshore *S. siderea* fragments exhibited higher physiological plasticity than inshore counterparts, suggesting that this offshore population was more susceptible to changing conditions. There were no plasticity differences in *P. strigosa* and *P. astreoides* between natal reef environments, however, temperature evoked stronger responses in both species. Interestingly, while each species exhibited unique physiological responses to ocean acidification and warming, when data from all three species are modelled together, convergent stress responses to these conditions are observed, highlighting the overall sensitivities of tropical corals to these stressors. Our results demonstrate that while ocean warming is a severe acute stressor that will have dire consequences for coral reefs globally, chronic exposure to acidification may also impact coral physiology to a greater extent in some species than previously assumed. Further, our study identifies *S. siderea* and *P. astreoides* as potential 'winners' on future Caribbean coral reefs due to their resilience under projected global change stressors, while *P. strigosa* will likely be a 'loser' due to their sensitivity to thermal stress events. Together, these species-specific responses to global change we observe will likely manifest in altered Caribbean reef assemblages in the future.

CoralPhysiology). Protocols for host carbohydrate and lipid assays can be accessed on protocols.io (carbohydrate: doi.org/10.17504/protocols.io. bvb9n2r6; lipid: doi.org/10.17504/protocols.io. bvcfn2tn).

**Funding:** This research was partially supported by the Women Diver Hall of Fame Sea of Change Foundation Marine Conservation Scholarship (https://www.wdhof.org/scholarship/marine-conservation-scholarship-graduate) and Lerner-Gray Memorial Fund of the American Museum of Natural History Grants for Marine Research (https://www.amnh.org/research/richard-gilder-graduate-school/academics-and-research/fellowship-and-grant-opportunities/research-grants-and-graduate-student-exchange-fellowships/the-lerner-gray-fund-for-marine-research) awarded to CBB. JBR acknowledges support from NSF BIO-OCE award #1437371 (https://www.nsf.gov/geo/oce/programs/biores. jsp). The funders had no role in study design, data collection and analysis, decision to publish, or preparation of the manuscript.

**Competing interests:** The authors have declared that no competing interests exist.

## Introduction

Human-induced global change is driving unprecedented challenges for ecosystems globally, from increases in terrestrial droughts [1] and severe storm activity across lower latitudes [2] to altering species' distributions globally [3,4]. Coral reefs are a prime example of an ecosystem heavily impacted by global change, particularly by ocean warming and acidification [5–8]. Ocean acidification and warming are predicted to affect many marine ecosystems by reducing ecosystem complexity and function, especially for organisms with longer generational times and thus fewer opportunities to adapt to changing conditions [9]. Therefore, understanding the diversity of responses of tropical reef-building corals at both the species- and population-levels is critical for predicting future impacts of global change.

Previous work assessing tropical reef-building corals under global change has generally focussed on quantifying changes in coral calcification rates owing to the ecological importance of new reef production for the maintenance of these ecosystems [10–15]. These studies demonstrate a diversity of calcification responses under stress, including maintained and suppressed growth rates [12,16,17]. For example, the Caribbean coral species *Siderastrea siderea* and *Porites astreoides* have been shown previously to maintain higher growth rates under ocean acidification and/or warming stress, [16–18] and other species, such as *Orbicella faveolata* and *Acropora cervicornis*, generally exhibited reduced growth under these same stressors [14,18,19]. While some corals sustain growth rates under stress, these corals may accomplish this at a cost to other metabolic processes [20,21] or through modifications to holobiont (animal host, dinoflagellates, bacteria, viruses, etc.) communities.

Tropical reef-building corals depend on the maintenance of an endosymbiotic relationship with photosynthetic dinoflagellates (family Symbiodiniaceae) for a significant portion of their energetic needs [22]. However, this relationship often breaks down under severe or prolonged stress, especially with increasing seawater temperature, resulting in the phenomenon known as 'coral bleaching' [23–25]. Corals bleach in response to ocean acidification, but especially in response to warming, and this loss of symbiosis leads to declines in calcification and gametogenesis [26]. Thus, as the symbiosis between the coral host and algal symbionts breaks down, both components of the coral are likely to exhibit closely integrated physiological responses. Indeed, previous work has observed that greater coral tissue biomass follows increased symbiont density and chlorophyll a content in several Caribbean reef-building coral species [27], highlighting the intrinsic relationship with algal symbionts to support the coral host's energy budget. Further, previous work has reported the influence of algal symbiont and microbiome communities as mechanisms of improving coral holobiont physiology under environmental stress [28–32]. Overall, it is clear that maintaining healthy symbioses within the coral holobiont is critical for the physiological health of the coral host.

Coral tissue biomass and energy reserves (e.g., lipid, protein, carbohydrate) are important aspects of overall coral health [33,34] that provide insight into resilience and recovery capacity in response to environmental stressors. Although energy reserves are extremely important in understanding the coral host response to stress, few studies have investigated how the combination of ocean acidification and warming influence these traits [34–36]. Coral tissue biomass relies on the equilibrium between energy sources and expenditures; thus, corals with already low biomass (i.e., low energy reserves) may experience heightened vulnerability under environmental stress [37] and may explain some of the variation of physiological responses to stress within and between species [16,17]. However, studies have demonstrated that corals may not always consume energy reserves under environmental stress [34] or increase metabolic processes [38]. Instead, corals may use other physiological mechanisms as coping tools to

maintain growth and host energy reserves, such as relying more on algal symbionts whose photosynthesis is fertilised under conditions of elevated $pCO_2$ [39].

Many symbiotic corals also have the capacity to exhibit physiological plasticity (i.e., modification of an organism's physiology) in response to changing environments that may be employed under global change scenarios [40,41]. While plasticity is often highlighted as a mechanism for rapid response to changing environments, there is debate about whether plasticity alone is enough to ensure species persistence under global change [42]. Indeed, a highly plastic coral may be able to modulate its physiology (e.g., increase chlorophyll a per symbiont cell) under an acute stress event (e.g., low light levels) [43], but this is likely to come at a cost to another metabolic process, such as energy stores. This physiological cost can be beneficial for the coral in the short term, however, may eventually result in a decline in fitness [42,44], especially in long-lived organisms like reef-building corals. These potential trade-offs in reef-building corals remain poorly understood and highlight the complexities of plasticity as a mode of global change resilience.

To assess the physiological responses of Caribbean corals to independent and combined ocean acidification (300–3290 µatm) and warming (28, 31°C), we conducted a 93-day common-garden experiment on 3 species of corals (*S. siderea*, *Pseudodiploria strigosa*, *P. astreoides*) and quantified coral host energy reserves (total protein, carbohydrate, lipid) and algal symbiont physiology (cell density, chlorophyll a concentration, coral colour intensity). These coral species were selected because they represent both weedy (*P. astreoides*) and stress-tolerant (*S. siderea* and *P. strigosa*) life histories [45], possess similar growth morphologies (mounding), and are common throughout the Caribbean across a variety of environmental gradients. Additionally, we included corals from two distinct reef environments to assess how environmental histories impact responses to global change stressors. Overall, we selected these species to better understand how corals that are expected to dominate Caribbean reefs in the future may respond to global change stressors. We hypothesised that (1) corals are more susceptible to thermal stress than acidification, (2) physiological responses are highly species-specific, and (3) physiological plasticity dictates coral resilience under global change. Our results highlight the diversity of physiological responses, from susceptibility to resistance, that Caribbean corals exhibit in response to projected global change, which will ultimately drive changes in community compositions across space.

## Methods

### Experimental design

Six colonies each of three Caribbean reef-building corals (*Siderastrea siderea*, *Pseudodiploria strigosa*, *Porites astreoides*) were collected from inshore (Port Honduras Marine Reserve; 16° 11'23.5314"N, 88°34'21.9360"W) and offshore (Sapodilla Cayes Marine Reserve; 16° 07'00.0114"N, 88°15'41.1834"W) reef environments at similar depths (3–5 m) from the southern portion of the Belize Mesoamerican Barrier Reef System. All corals were collected following local laws and regulations with appropriate permits (#5674). These two distinct reef environments are approximately 25 km apart and were selected to explore how environmental history (e.g., temperature, salinity, carbonate chemistry, nutrients, etc.) affects responses to global change. Specifically, the inshore site is known to be more environmentally variable (i.e., diel and seasonal variability) than the offshore location (**S1 Fig**), potentially diving local adaptation and/or long-term acclimatisation in these species [46–48]. This study further investigates the physiological responses of corals assessed in Bove et al. [17] and detailed descriptions of experimental setup can be found there.

Corals (2 reef environments x 3 species x 6 colonies = 36 colonies) were collected and transported to Northeastern University's Marine Science Center. Colonies were sectioned into eight equally-sized fragments (8 fragments/colony = 288 total samples) and returned to ambient conditions for a 23-day recovery period, followed by a 20-day acclimation period where tanks were slowly adjusted to target experimental treatment conditions. Corals were maintained in one of eight experimental treatments (three replicate tanks per treatment; see **S2 Fig** for coral allocation schematic and **Table A in S1 Text** for sample sizes) for the 93-day experiment. The eight treatments encompassed four $pCO_2$ treatments (Table 1) corresponding to pre-industrial, current-day ($pCO_2$ control), moderate end-of-century, and an extreme $pCO_2$ level all crossed with two temperatures (Table 1) corresponding to the corals' approximate present-day summer mean and projected end-of-century summer warming[49] that has also been observed to induce bleaching in these species [50]. High-precision digital solenoid-valve mass flow controllers (Aalborg Instruments and Controls; Orangeburg, NY, USA) were used to bubble air alone (control $pCO_2$ conditions), or in combination with $CO_2$-free air (pre-industrial conditions) or $CO_2$ gas (end-of-century and extreme conditions) to achieve gas mixtures of each desired $pCO_2$ condition.

Experimental tanks were filled with 5 μm-filtered natural seawater from Massachusetts Bay with a salinity of 31.7 psu (±0.2) and were illuminated with full spectrum LED lights on a 10:14 light-dark cycle with photosynthetically active radiation of approximately 300 μmol photons m$^{-2}$ s$^{-1}$. Corals were fed a combination of ca. 6 g frozen adult Artemia and 250 mL concentrated newly hatched live Artemia (500 mL-1) every other day to satisfy heterotrophic feeding [51,52]. Temperature, salinity and pH were measured at the same time (~1PM) every other day throughout the experiment and total alkalinity (TA) and dissolved inorganic carbon (DIC) were analysed every 10 days with a VINDTA 3C (Marianda Corporation, Kiel, Germany) (**S3 Fig**). Temperature, salinity, TA, and DIC were used to calculate carbonate parameters using $CO_2$SYS [53] with Roy *et al.* [54] carbonic acid constants $K_1$ and $K_2$, Mucci's value for the stoichiometric aragonite solubility product [55], and an atmospheric pressure of 1.015 atm. At the completion of the experimental period, corals were immediately flash-frozen in liquid nitrogen and transported to the University of North Carolina at Chapel Hill. Coral tissue was removed from the skeleton using seawater with an airbrush and stored in 50 mL conical tubes at −80˚C until further processing.

## Host and symbiont physiological parameter assessments

Preserved coral tissue slurries were homogenised with a *Tissue-tearor* (BioSpec Products; Bartlesville, Oklahoma, USA) for several minutes and vortexed for 5 seconds, after which 1.0 mL of slurry was aliquoted for algal symbiont density analysis. Algal symbiont aliquots were dyed with 200 μL of a 1:1 Lugol's iodine and formalin solution and cell densities were quantified by performing at least 3 replicate counts of 10 μL samples using a hemocytometer (1 x 1 mm; Hausser Scientific, Horsham, Pennsylvania, USA) and a compound microscope. Algal symbiont densities were standardised to total tissue volume and previously measured coral surface area ($10^6$ cells per cm$^2$) [17]. Remaining tissue slurry was centrifuged at 4400 rpm for 3 minutes to separate the coral host and algal symbiont fractions, and the host fraction was poured

**Table 1. Warming and acidification treatment means and standard deviations.**

| Temperature Treatment (˚C) | | $pCO_2$ treatment (μatm) | | | |
|---|---|---|---|---|---|
| | | **Pre-industrial** | **Current-day** | **End-of-century** | **Extreme** |
| **Control** | 28 ± 0.4 | 288 ± 65 | 447 ± 152 | 673 ± 104 | 3285 ± 484 |
| **Warming** | 31 ± 0.4 | 311 ± 96 | 405 ± 91 | 701 ± 94 | 3309 ± 414 |

off from the symbiont pellet. Chlorophyll a pigment was extracted from the algal pellet by adding 40 mL of 90% acetone to the conical tube at −20˚C for 24 hours. Samples were diluted by adding 0.1 mL of extracted chlorophyll a sample to 1.9 mL of 90% acetone. If samples were too high or too low for detection on the fluorometer, samples were reanalysed by either diluting or concentrating the sample, respectively. Extracted chlorophyll a content was measured using a Turner Design 10-AU fluorometer with the acidification method [56] and expressed as the µg of pigment per $cm^2$ of coral tissue surface area.

Coral host supernatant was aliquoted (1 mL each) for total protein, carbohydrate, and lipid analysis, and stored at −80˚C. Glass beads were added to total protein aliquots, vortexed for 15 minutes, and centrifuged for 3 minutes at 4000 rpm. Duplicate samples were prepared with 235 µL of seawater, 15 µL of protein aliquot, and 250 µL of Bradford reagent (*Thermo Scientific*) and left for 20 minutes. Coral host total protein samples were read at 562 nm on a spectrophotometer (Eppendorf BioSpectrometer® basic; Hamburg, Germany) in duplicates and were expressed as mg per $cm^2$ coral tissue surface area. For coral host carbohydrate, 25 µL of phenol was added to 1000 µL of diluted coral host slurry and vortexed for 3 seconds before immediately adding 2.5 mL concentrated sulphuric acid ($H_2SO_4$). Samples were incubated at room temperature for 1 minute and then transferred to a room temperature water bath for 30 minutes [57]. Finally, 200 µL of each standard and sample was pipetted into a 96-well plate in triplicate and read on a spectrophotometer at 485 nm (BMG LABTECH POLARstart Omega; Cary, North Carolina, USA). Total carbohydrate was expressed as mg per $cm^2$ coral tissue surface area [58]. Coral host lipids were extracted following the Folch Method [59] by adding 600 µL of chloroform ($CHCl_3$) and methanol ($CH_3OH$) in a 2:1 ratio to 600 µL of host slurry and placed on a plate shaker for 20 minutes before adding 160 µL of 0.05M sodium chloride (NaCl). Tubes were inverted twice and then centrifuged at 3000 rpm for 5 minutes. Finally, the lipid layer was removed and 100 µL was pipetted in triplicate into a 96-well plate for colourimetric assay. The lipid assay was performed by adding 50 µL of $CH_3OH$ to each well before evaporating the solvent at 90˚C for 10 minutes. Next, 100 µL of $H_2SO_4$ was added to every well, incubated at 90˚C for 20 minutes, and cooled on ice for 2 minutes before transferring 75 µL of each sample into a new 96-well plate. Background absorbance of the new plate was read at 540 nm on a spectrophotometer before adding 34.5 µL of 0.2 mg/mL vanillin in 17% phosphoric acid to each well. The plate was read again at 540 nm and coral host lipid concentrations were normalised to coral surface area (mg per $cm^2$) [60,61].

Coral colour intensity was also analysed from images of every fragment with standardized colour scales taken at every 30 days throughout the experiment. This assessment complements other algal symbiont physiological assessments as a non-destructive alternative to quantify coral bleaching. Colour balance was adjusted using a custom Python script that took a square of pixels as a white standard (50 x 50) on each image to adjust the colour balance until it was true white. The total red, green, blue, and sum of all colour channel intensities were measured following [62] using the MATLAB macro "AnalyzeIntensity" for either 10 (*S. siderea* and *P. astreoides*) or 20 (*P. strigosa*; 10 in valley and 10 on ridges) quadrats of 25 x 25 pixels on each coral fragment. The resulting values act as a measure of brightness, with higher brightness values correlating with pigment lightening (i.e., coral bleaching); thus, data were inverted so that lower values represent reduced coral pigmentation. The sum of all colour channels (red, green, blue) resulted in a stronger correlation with symbiont physiology (chlorophyll a and cell density) in *S. siderea* and *P. strigosa*, while the red channel alone was best in *P. astreoides*.

## Coral physiology analyses

Sample mortality was observed throughout the experimental period across species as described in Bove et al. [17] and thus some treatments resulted in reduced replication for physiological

analyses (**Table A in S1 Text**). Overall, *S. siderea* exhibited nearly 90% survival (86 total fragments), *P. strigosa* exhibited 80% survival (77 total fragments), and *P. astreoides* exhibited 72% survival (69 total fragments) at the end of the experiment [17]. Further, the initial and final control treatment sample size of *P. strigosa* was lower than other species because this treatment system had to be reconstructed before the start of the experiment and there were only a few reserve genotypes of this species available for the new control system.

Principal component analysis (PCA) (function *prcomp*) of scaled and centered physiological parameters (host carbohydrate, host lipid, host protein, algal symbiont chlorophyll a, algal symbiont cell density, calcification rate as previously reported for the same samples in Bove et al [17]) were employed to assess the relationship between physiological parameters and treatment conditions for each coral species. Main effects (temperature, $pCO_2$, reef environment) were evaluated with PERMANOVA using the *adonis2* function (*vegan* package; version 2.5.7 [63]). The additive model resulted in a lower AIC than the fully interactive model for all species, so interaction terms were dropped from each model resulting in fully additive models (see **Table B in S1 Text**).

Correlations of all physiological parameters were assessed to determine the relationships between parameters within each species. The Pearson correlation coefficient ($R^2$) of each comparison was calculated using the *corrgram* package (version 1.13 [64]) and the significance was calculated using the *cor.test* function. These relationships were then visualised through simple scatterplots.

Here, we use physiological plasticity to refer to the amount an individual modifies its physiology in response to stress compared to observed physiology under control conditions. Physiological plasticity of each experimental fragment was calculated for each species using all principal components (PCs) calculated above as the distance between an experimental fragment and the control (420 μatm; 28˚C) fragment from that same colony [65]. The effects of treatment ($pCO_2$ and temperature) and natal reef environment on calculated distances were assessed using generalised linear mixed effects models (function *lmer*) with a Gamma distribution and log-link and a random effect for colony (*P. strigosa* and *P. astreoides*) or tank crossed with colony (*S. siderea*). The best-fit model was selected as the model with the lowest AIC for each species (**Table C in S1 Text**). Natal reef environment was only a significant predictor of plasticity in *S. siderea*; thus, samples were pooled across reef environments for both *P. strigosa* and *P. astreoides*. Parametric bootstraps were performed to model mean response and 95% confidence intervals with 1500 iterations and significant effects were defined as non-overlapping confidence intervals. Marginal and conditional $R^2$ values of the best fit models were calculated using the *r2_nakagawa* function in the rcompanion package (version 2.4.13 [66]). All figures and statistical analyses were carried out in R version 4.1.2 (R Core Team, 2018) and the accompanying data and code can be freely accessed on GitHub (github.com/seabove7/Bove_CoralPhysiology) and Zenodo (10.5281/zenodo.5093907).

## Results

### Principal component analysis

Two PCs explained approximately 66% of the variance in physiological responses of *S. siderea* to ocean acidification and warming treatments (**Fig 1A**). PC1 was driven by differences in algal symbiont physiology (chlorophyll a, cell density), while PC2 represented an inverse relationship between host energy reserves (lipid, protein, carbohydrate) and calcification rates and colour intensities. Overall, higher $pCO_2$ and temperature resulted in reduced *S. siderea* physiology (**Fig 1A**). Treatment $pCO_2$ predominantly drove *S. siderea* physiological responses (p = 7e-04), while temperature and reef environment did not explain as much variation in physiological responses

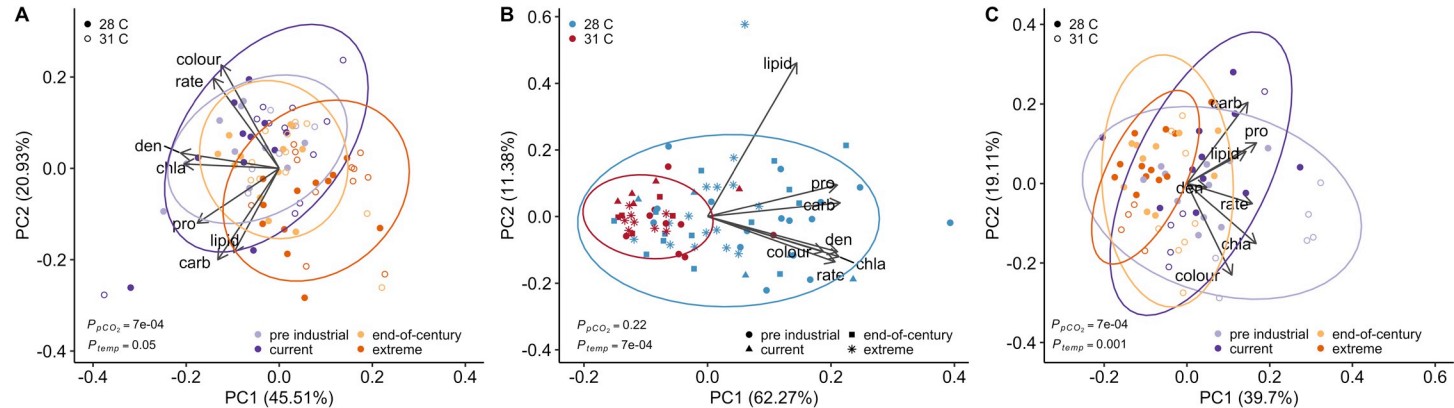

**Fig 1.** Principal component analysis (PCA) of all coral physiological parameters for (**A**) *S. siderea*, (**B**) *P. strigosa*, and (**C**) *P. astreoides* after 93 days of exposure to different temperature and $p$CO$_2$ treatments. PCAs of (**A**) *S. siderea* and (**C**) *P. astreoides* are depicted by $p$CO$_2$ in colour (pre industrial [300 μatm], light purple; current day [420 μatm], dark purple; end-of-century [680 μatm], light orange; extreme [3290 μatm], dark orange) and temperature by shape (filled circles 28°C; open circles 31°C). The PCA for (**B**) *P. strigosa* is depicted by temperature in colour (28°C blue; 31°C red) and $p$CO$_2$ by shape (pre industrial, circles; current day, triangles; end-of-century, squares; extreme, stars). Arrows represent significant (p < 0.05) correlation vectors for physiological parameters (rate = calcification rate; den = symbiont density; chla = chlorophyll a; pro = protein; carb = carbohydrate; lipid = lipid; colour = colour intensity) and ellipses represent 95% confidence based on multivariate t-distributions.

(p = 0.05 and p = 0.001, respectively; **Table D in S1 Text** and **S4A Fig**). These observed responses are driven by declines in total host physiology under warming as well as reduced symbiont physiology with increasing $p$CO$_2$ (**S5A Fig**). Further, no significant interactive effect between temperature and $p$CO$_2$ was detected in *S. siderea* physiology (**S4D Fig**).

For *P. strigosa*, 74% of the variance in response to treatments was explained by two PCs (**Fig 1B**). PC1 explained most of the variation of physiological parameters with the exception of host lipid content, which was represented in PC2. Physiology of *P. strigosa* was reduced under warming (p = 7e-04) and in offshore samples (p = 7e-04; **S4B Fig**), however, $p$CO$_2$ did not clearly alter physiology (**Fig 1B**; p = 0.2; **Table D in S1 Text**). This clear decline in physiology under warming is driven by declines in symbiont physiology and total host protein content (**S5B Fig**). Again, no significant interactive effect between temperature and $p$CO$_2$ was detected (**S3E Fig**).

For *P. astreoides*, the first two PCs explained 59% of the total variance in response to treatment (**Fig 1C**). Samples separated most along PC1 driven primarily by calcification rate and algal symbiont density, while PC2 exhibited an inverse relationship between host total carbohydrate and colour intensity. Overall, higher $p$CO$_2$ reduced *P. astreoides* physiology, while elevated temperature resulted in improved physiology (**Fig 1C**). These patterns are most notable in the reduced host energy reserves in response to increasing $p$CO$_2$ and higher symbiont physiology and lipid content under warming (**S5C Fig**). Temperature (p = 0.001) and $p$CO$_2$ (p = 7e-04) altered *P. astreoides* physiology, while reef environment was not significant (p = 0.5; **Table D in S1 Text** and **S4C Fig**) and there was no significant interactive effect between temperature and $p$CO$_2$ (**S3F Fig**).

## Correlations of physiological parameters

Coral physiological parameters were generally positively correlated with one another within each of the three species. Correlations between *S. siderea* physiological parameters identified 15 significant relationships out of all 21 possible comparisons (**Fig 2A**). Of those significant correlations, six resulted in a Pearson's correlation coefficient ($R^2$) equal to or greater than 0.5, with the strongest relationship identified between symbiont density and chlorophyll a ($R^2$ = 0.72).

All pairwise physiological parameters were significantly correlated with one another in *P. strigosa* and, of those, 15 correlations exhibit moderate ($R^2 > 0.50$) positive relationships (**Fig 2B**). Notably, the two strongest correlations were host carbohydrate vs. host protein ($R^2 = 0.70$) and host carbohydrate vs. chlorophyll a ($R^2 = 0.76$).

Compared to both *S. siderea* and *P. strigosa*, fewer physiological traits were significantly ($p < 0.05$) correlated with one another in *P. astreoides* (12 significant out of 21 total comparisons; **Fig 2C**). Of the significant correlations, only two pairwise comparisons resulted in a Pearson's correlation coefficient greater than 0.5: chlorophyll a vs. colour intensity ($R^2 = 0.57$) and host carbohydrate vs. host protein ($R^2 = 0.68$).

## Coral physiological plasticity

Physiological plasticity of offshore *S. siderea* fragments exhibited a positive linear trend with increasing $pCO_2$, while the inshore fragments appear to respond in a parabolic pattern to $pCO_2$, with the lowest calculated distances occurring at 420 µatm, 31°C and 680 µatm, 28°C (**Fig 3A**). Further, offshore *S. siderea* fragments exhibited higher plasticity in the extreme $pCO_2$ treatment than in inshore fragments reared in the pre-industrial, current-day, and extreme $pCO_2$ treatments, regardless of temperature (**Fig 3A and Table E in S1 Text**).

Plasticity of *P. strigosa* and *P. astreoides* was not clearly different between colonies based on natal reef environments (see **Table C in S1 Text**). No clear differences in physiological plasticity in response to treatment were identified in *P.strigosa* (**Fig 3B and Table E in S1 Text**), however, this is likely due to reduced sample sizes in this analysis as a result of only five colonies ($N_{offshore} = 3$, $N_{inshore} = 2$) present in the control treatment for distance calculations.

Elevated temperature generally resulted in higher plasticity of *P. astreoides* compared to control temperature (**Fig 3C and Table E in S1 Text**), however, this trend was not clearly different within each $pCO_2$ treatment. Physiological plasticity of *P. astreoides* was significantly lower in both the pre-industrial and end-of-century $pCO_2$ treatments at control temperature than that measured in the extreme $pCO_2$ treatment combined with the elevated temperature.

## Species differences in coral physiology

The first two PCs of coral physiology explained about 62% of the total variance across samples (**Fig 4**). In general, fragments of *S. siderea* contained higher chlorophyll a content, host

**A)** *S. siderea*

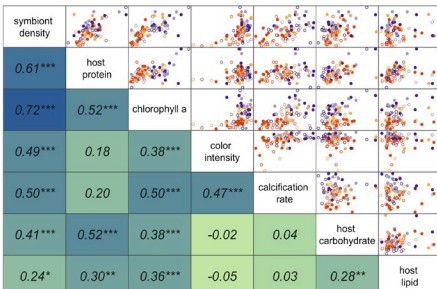

**B)** *P. strigosa*

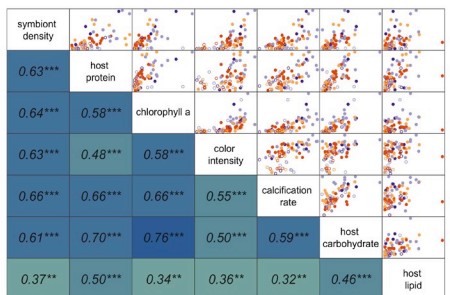

**C)** *P. asteroides*

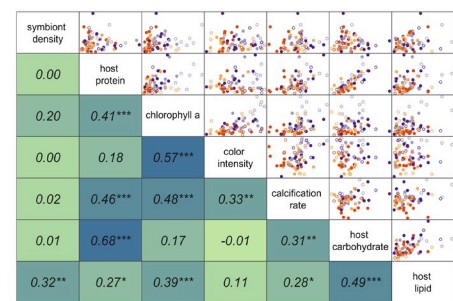

**Fig 2.** Coral physiological parameter scatter plots (top) and correlation matrices (bottom) for (**A**) *S. siderea*, (**B**) *P. strigosa*, and (**C**) *P. astreoides* showing pairwise comparisons of within each species. Scatter plots of each pairwise combination of physiological parameters are displayed on the top with temperature treatment depicted by shape (28°C closed points; 31°C open points) and $pCO_2$ treatment depicted by colour (pre industrial [300 µatm], light purple; current day [420 µatm], dark purple; end-of-century [680 µatm], light orange; extreme [3290 µatm], dark orange). Strengths of the correlations ($R^2$ via Pearson correlation coefficients) between each pairwise combination of physiological parameters are indicated by darker shades of blue on the bottom with significance depicted by asterisks according to significance level (* $p < 0.05$; ** $p < 0.01$; *** $p < 0.001$). $R^2$ and significance levels correspond to the scatter plot at the intersection between two physiological parameters.

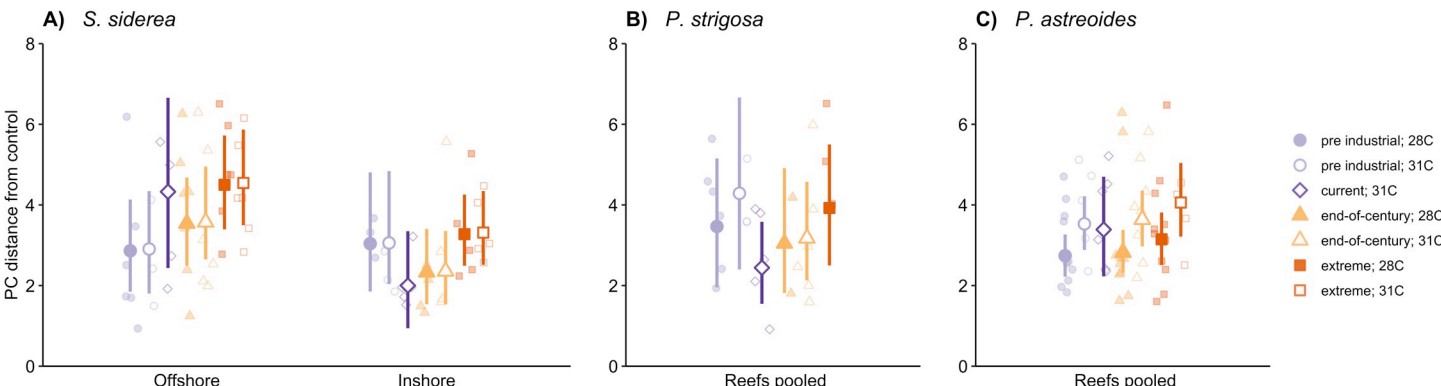

**Fig 3.** Physiological plasticity of (**A**) *S. siderea*, (**B**) *P. strigosa*, and (**C**) *P. astreoides* after 93-day exposure to experimental treatments. Higher values represent greater plasticity (stronger response) in coral samples. Natal reef environment is depicted along the x axis for *S. siderea*, however, *P. strigosa* and *P. astreoides* samples were pooled by reef environment. $pCO_2$ treatment is depicted by colour and shape (pre industrial [300 μatm], light purple; current day [420 μatm], dark purple; end-of-century [680 μatm], light orange; extreme [3290 μatm], dark orange) and temperature is represented as either closed (28°C) or open (31°C) symbols. The current day at 28°C treatment is not depicted here since plasticity is represented as the distance from this treatment (420 μatm at 28°C). Symbols and bars indicate modelled means and 95% confidence intervals. Non overlapping confidence intervals were interpreted to be statistically different.

carbohydrate, and host lipid content, while *P. strigosa* fragments typically had greater host protein content accompanied by higher calcification rates, and fragments of *P. astreoides* were differentiated by their high symbiont densities (**Figs 4A and S6**). Despite being different coral species, coral physiology exhibited similar declines in responses to increasing $pCO_2$ treatments (**Fig 4B**), however, responses to temperature were highly species-specific (**Fig 4C and Table F in S1 Text**). Furthermore, corals from the inshore reef environment exhibited more constrained physiology than their offshore counterparts (**S6 Fig**).

## Discussion

### Coral physiology highlights sensitivity of Caribbean corals to global change

Caribbean coral reefs have experienced considerable shifts in ecosystem composition since the 1970s defined by declines in several stony coral taxa [67,68], resulting in reefs now dominated by weedy and stress-tolerant species. Ocean acidification, warming, and the combination of

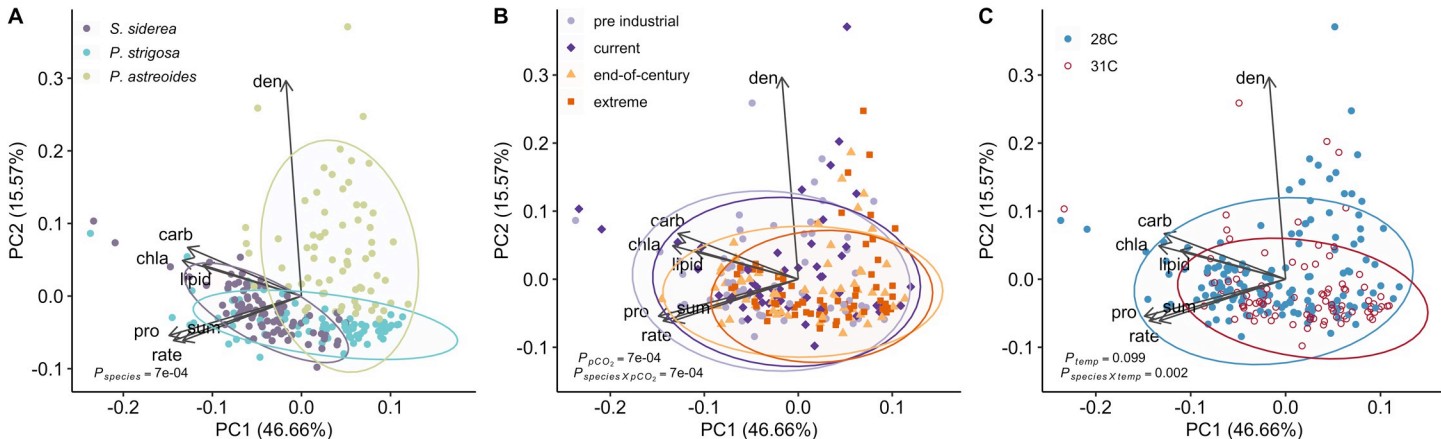

**Fig 4.** Principal component analysis (PCA) comparing the physiology of all three species at the end of the experiment with samples clustered by (**A**) species, (**B**) $pCO_2$ treatment, and (**C**) temperature treatment. Arrows represent significant (p < 0.05) correlation vectors for physiological parameters and ellipses represent 95% confidence based on multivariate t-distributions.

the two stressors are expected to further reduce coral abundance throughout the Caribbean by the end of this century [69]. We demonstrate a variety of coral responses to simulated ocean acidification and warming scenarios that provide insight into how multiple stress-tolerant and weedy coral species may respond to global change. Understanding individual physiological responses of coral hosts and their algal symbionts provides valuable insight into the relationship between these partners, especially in these now-dominant species. However, to better predict how corals will respond to global change, it is necessary to assess how the physiological parameters of both partners will respond. For example, we found that $pCO_2$ treatment drove differences in coral physiology of both *S. siderea* and *P. astreoides* (**Fig 1**); however, these effects were not clear when assessing individual physiological parameters on their own within a species (**S5 Fig**). Indeed, several previous studies have reported mixed physiological responses to elevated $pCO_2$, from no difference in coral host energy reserves [34] to reduced symbiont density and productivity loss [25,38]. These effects of $pCO_2$ highlight the complexity of the responses of corals under stress [34,70,71] and suggest that, by limiting assessments to only a few physiological parameters, studies may miss important changes to the coral's overall condition.

Coral physiologies of all three species were also modulated by temperature, although these impacts were more variable. *Siderastrea siderea* and *P. strigosa* both exhibited declines in physiology under elevated temperature (31˚C) (**Figs 1** and **S5**); however, these declines were more pronounced in *P. strigosa*, especially through time (**Figs 1B** and **S7**). Indeed, while *P. strigosa* was previously classified as a stress-tolerant species based on trait assessment [45], it has more recently been identified as a more thermally sensitive coral species [35,72,73]. This response is likely representative of the overall deterioration of coral condition in response to thermal stress, which may lead to mortality under chronic or extreme exposure as is being seen more frequently on Caribbean coral reefs [5]. Thermal events on coral reefs are generally considered to be acute stress events (on the scale of hours to weeks) [74]. Thus, exposure of these corals to more than 90 days of constant elevated temperature may have elicited a more severe response in *P. strigosa* as is seen during mass bleaching events *in situ* for this species [75]. Conversely, elevated temperature corresponded with improved physiological parameters in *P. astreoides*. These differences in coral thermal responses are not surprising given that *P. astreoides* is generally considered a more opportunistic coral that can persist in less-desirable conditions, including elevated temperature [18,45,76]. Conversely, *S. siderea* and *P. strigosa* are classified as 'stress tolerant' species with varying levels of susceptibility and resilience to environmental stress [16,50,77,78]. Despite some similarities in responses to ocean acidification and warming observed here, the different relationships between physiological parameters within each species likely interact to produce the species-specific responses observed *in situ*.

A major goal of this study was to better understand the combined effects of ocean warming and acidification on coral physiology since these stressors continue to change in lock step. While many studies report synergistic effects (i.e., the effects of both stressors compounding one another) of increasing temperature and $pCO_2$ on coral responses [79–81], the interaction term of these treatments in our study was not significant in any models performed. In fact, the species assessed in our experiments generally exhibited clear responses to either warming (*P. strigosa*) or acidification (*S. siderea* and *P. astreoides*) that was only exacerbated by the other stressor in the high temperature, extreme acidification scenario (**S4 Fig**). Under the combined acidification and warming scenarios, it is possible that one stressor counteracted the effects of the other to result in marginal physiological changes [82]. Indeed, it has been suggested that $CO_2$ fertilisation of algal symbionts under ocean acidification may improve coral physiology [39], potentially countering the negative effects of associated warming on coral-algal symbiosis. Conversely, metabolic processes generally improve along with increasing temperatures up

to an individual's thermal optimum [83], suggesting that the elevated temperature used here may have supported improved physiology, counteracting any negative effects of ocean acidification. Further, while other studies report synergistic effects on coral physiology, most of these studies only assess a single parameter, potentially missing other key physiological responses that suggest more additive responses like observed here. It is clear that coral responses under global change remain complex and require further investigation using additional multi-stressor, multi-species studies to tease apart these complexities.

## Global change and species-specific drivers of physiological plasticity

On shorter ecological time scales–like those employed in this experiment–plasticity may be a coral's most efficient response to global change, as it permits individual-level acclimatisation to a rapidly changing environment within a generation [40,42]. Plasticity has been identified as an important mechanism in coping with elevated $pCO_2$ conditions in tropical corals [84–86] and may predict how these organisms will perform under global change. However, physiological plasticity may not always be beneficial long term and may instead signal a shift in organism condition [18,42,44]. Organisms exhibiting higher plasticity in response to environmental change (e.g., ocean warming and acidification) may incur a physiological cost in the form of a trade-off that ultimately may impact the population's ability to resist future change [40–42]. Here we assessed the physiological plasticity of the coral under elevated temperature and $pCO_2$, and compared these responses across two natal reef environments (inshore vs. offshore). We found that *S. siderea* fragments from the offshore exhibited higher plasticity in response to extreme $pCO_2$ (3290 μatm) compared to the inshore counterparts, and that this pattern differed between the two habitats (**Fig 3A**). These results suggest that offshore *S. siderea* fragments modulated their physiology to a greater extent than the inshore corals and this shift in physiological state suggest reduced capacity to persist under future ocean acidification. This higher plasticity likely comes at a fitness trade-off in corals that are experiencing sub-optimum conditions [42,87]. Indeed, a reciprocal transplant experiment in southern Belize identified higher plasticity of offshore colonies of *S. siderea* compared to those from a nearshore environment [47]. The offshore colonies grew at a much higher rate when transplanted to the nearshore environment than in their natal environment (generally considered more ideal conditions) [47], suggesting that plasticity in these corals may indeed come at the cost of growth in home or more ideal conditions [42].

Varying levels of plasticity in *P. strigosa* and *P. astreoides* from different habitats has been previously reported [47,88]; however, natal reef effects were not evident in either species in this study (**Fig 3B and 3C**). The small sample size of *P. strigosa* likely contributed to the lack of differences between habitats, while different measures of plasticity–physiological plasticity (present study) vs. gene expression plasticity [88]–may contribute to the inconsistent responses observed in *P. astreoides*. While neither species exhibited differing levels of plasticity between reef environments, both *P. strigosa* and *P. astreoides* appear to exhibit higher plasticity at the elevated temperature, though this is only statistically significant in *P. astreoides* (**Fig 3B and 3C**). Interestingly, the higher plasticity at elevated temperatures in *P. strigosa* was associated with diminished physiological conditions, while higher plasticity in *P. astreoides* manifested as improved physiology (**Fig 1B and 1C**). These differences highlight how plasticity may result from physiological trade-offs in response to environmental change in some organisms (i.e., *P. strigosa*) [42,87], while other organisms (i.e., *P. astreoides*) may benefit from such plastic responses to match their physiology to their environment [89]. Either way, the role of plasticity in coral responses to global change is complex and merits further investigation to better understand species-specific levels of resilience.

Another explanation for varying susceptibilities across coral species under global change may relate to how physiological parameters are correlated to one another within the coral. For example, all physiological parameters were significantly correlated with one another for *P. strigosa* (**Fig 2B**), while only some correlations were significant for *S. siderea* and *P. astreoides* (**Fig 2**). Notably, while symbiont density was significantly correlated with all parameters in *P. strigosa*, it was least correlated with host lipid content, which was in turn best correlated with host protein and host carbohydrate (**Fig 2B**). This pattern suggests *P. strigosa* are consuming carbohydrate and protein stores in response to reduced symbiont density and chlorophyll a content, while lipid stores remain relatively unaltered, in line with previous work on coral energetics [90,91]. *Siderastrea siderea* exhibited similar relationships between symbiont density and all other physiological parameters; however, calcification rates were more dependent on algal symbiont status than host energy reserves (**Fig 2A**). Interestingly, *P. astreoides* symbiont density only resulted in a significant correlation with lipid content, while chlorophyll a was a better predictor of most physiological parameters (**Fig 2C**). In fact, chlorophyll a and symbiont density resulted in one of the strongest correlations in both *S. siderea* and *P. strigosa*, while these two parameters were not correlated in *P. astreoides*. This suggests that *S. siderea* and *P. strigosa* both rely on greater concentrations of algal symbionts with higher chlorophyll a content for autotrophically-derived carbon to support the coral host [22,92], while *P. astreoides* is dependent on more efficient symbionts alone [93,94]. Additionally, these three species are known to host varying algal symbiont communities (e.g., *Siderastrea siderea* predominantly hosts *Cladocopium*; *P. strigosa* hosts *Cladocopium* and *Breviolum*; *P. astreoides* hosts *Breviolum* and *Symbiodinium* [95,96]) that may determine differing carbon allocation to the host as well as different thermal tolerances of the coral [97,98]. Although profiling of the algal symbiont community was outside the scope of the current study, both temperature and $p$CO$_2$ can modulate the symbiosis between coral hosts and algal symbionts [24,25,99,100]. Therefore, given that algal symbiont community and physiology play a significant role in coral responses to global change stressors, these types of data should be obtained in future experiments to better understand differences between and within tropical coral species.

Interestingly, when comparing PCAs of physiology from host only (lipid, carbohydrate, protein) and symbiont only (chlorophyll a, symbiont density, colour intensity) for each species, algal symbionts were generally more impacted than hosts by increasing $p$CO$_2$ (i.e., $p$CO$_2$ significantly drove differences in physiology in algal symbionts, not coral hosts) (**S8**–**S10 Figs** and **Table G in S1 Text**). For example, variance in *S. siderea* host physiology was not significantly explained by $p$CO$_2$; however, $p$CO$_2$ altered symbiont physiology. This result suggests that algal symbiont traits were being negatively impacted under ocean acidification, but that host energy reserves remained unaffected. This pattern contrasts previous work demonstrating no change in symbiont physiology under increased $p$CO$_2$ [34,101,102] and others highlighting greater transcriptomic plasticity of coral hosts in response to increasing $p$CO$_2$ relative to their algal symbionts [103]. Davies et al., [103] interpreted this result as the coral host responding poorly to $p$CO$_2$ stress. However, our results suggest that coral hosts were able to maintain energy reserves despite reductions in symbiont density and chlorophyll a content. There is debate on the exact relationship between the coral host and algal symbionts (i.e., mutualism vs. parasitism) as well as their relative roles in coral bleaching [104–106]. While this symbiotic relationship is largely considered a mutualism, recent work has highlighted that this relationship is context dependent and, under specific circumstances, the algal symbionts may become more parasitic [107]. Regardless, it is clear that understanding the varied responses of the different symbiotic partners is critical for predicting the future of tropical coral reefs.

### Global change drives similar physiological responses in Caribbean corals

Our results indicate species-specific relationships between physiological parameters within a coral that dictate responses to global change stressors and these patterns may separate the 'winners' from 'losers' on future reefs [108,109]. Comparisons across all experimental coral fragments highlight that *S. siderea* were differentiated by their higher host carbohydrate, host lipid, and chlorophyll a content, while *P. strigosa* fragments were associated with higher host protein and net calcification rates, and *P. astreoides* hosted the highest algal symbiont densities (**Fig 4A**). These physiological differences across species likely correspond to species-specific responses observed in this study and previous work assessing global change on tropical reef-building corals [16,17,48,110], as well as patterns of resilience observed *in situ* [76,78]. For example, *S. siderea* has generally been considered a more resilient species in terms of survival and growth when reared under ocean acidification and warming conditions [16,17,48]. This resilience may be associated with this species' maintenance of higher host carbohydrate reserves as a result of greater chlorophyll a content [111] along with increased host lipids reserves for long-term performance [90,91]. The association of proteins with *P. strigosa* is also noteworthy given that corals generally obtain proteins from their algal symbionts [112]. However, *P. strigosa* was the most bleached of the three species (see **S5** and **S7 Figs**), suggesting that this species exhibited the largest variation in protein as a result of the loss of productive symbionts with warming. These differences across species not only highlight differences in the underlying response strategies of Caribbean coral species, but may also assist in predicting responses to environmental stress.

Although the coral species examined here exhibit differing host and symbiont physiological responses, patterns of coral physiology converge under increasing $pCO_2$, but not elevated temperature, regardless of species (**Fig 4B and 4C**). This pattern observed with increasing $pCO_2$ cautions that the broad classification of coral species as 'resistant' or 'susceptible' to environmental stressors based on individual physiological responses [16,17,34,45,113] may overgeneralize sensitivity to future reef projections [6,16,69]. For example, recent observations of reduced recruitment and size distributions of *P. astreoides*, commonly labelled a 'winning' coral species across the Caribbean [114], suggest that qualifying the success of a species based on short-term studies or limited data (e.g., only measuring a single response parameter) may misrepresent its long-term trajectory. We are already witnessing species that were previously classified as stress-tolerant (i.e., *P. strigosa*) [45] shifting into a more susceptible category in the past several years alone [17,72], further highlighting the need to reassess how we label resilience in tropical reef-building corals. Similarly, Caribbean coral reef communities have experienced dramatic shifts in species composition and abundance over the past several decades [68]; therefore, many of the individuals within a species assessed today remain due to some level of resilience to stress. Overall, the susceptibility observed in this study across all species is indicative of future Caribbean coral reef assemblages composed only of the most tolerant individuals within a species, despite some species-level resilience to global change stressors.

## Conclusions

As global change continues, it is critical to understand species-specific responses to ocean acidification and warming scenarios to predict the future of Caribbean reef assemblages, especially with a focus on now-dominant coral species explored here. Our results suggest that *S. siderea* may continue to dominate reefs across the Caribbean due to its maintenance of tissue energy reserves and relatively unaltered symbiosis with their algal symbionts under stress. Conversely, the previously assumed stress-tolerant species *P. strigosa* was unable to maintain any physiological traits under warming, suggesting that this species is now particularly vulnerable to

thermal stress, which will likely lead to widespread bleaching and mortality. Finally, *P. astreoides* exhibited improved physiology under warming while ocean acidification caused reductions in the same physiological traits, indicating that this species may also fare better than others under global change. Although these species had variable responses under these global change scenarios, all three exhibited physiological deterioration under the effects of ocean acidification. Our results underscore the intricacies of coral physiology, both within and across species, in response to their environment and contribute to our understanding of the many ways that global change affects tropical coral reefs.

## Supporting information

**S1 Fig. *In situ* satellite sea surface temperature.** Monthly MODIS satellite SST data from 2002 to 2021 for both the inshore (Port Honduras Marine Reserve; yellow) and the offshore (Sapodilla Cayes Marine Reserve; green) coral collection locations. Solid horizontal lines represent corresponding reef environment mean SST across duration. The blue dashed line represents the experimental control treatment temperature (28 C) and the red dashed line represents the experimental elevated temperature treatment (31 C). Note the temperature variability of the inshore site exceeding the offshore location. [Data accessibility: *NASA OBPG. 2020. MODIS Aqua Global Level 3 Mapped SST. Ver. 2019.0. PO.DAAC, CA, USA. Dataset accessed [2021-02-02] at https://doi.org/10.5067/MODSA-MO4D9*].
(TIFF)

**S2 Fig. Experimental design layout.** Diagram showing allocation of coral fragments for a single species throughout the experiment. Colour represents a different colony and shape represents reef environment. Four colonies (two from each reef environment) are reared within each tank (grey box), with three tanks comprising a treatment (white box). This is repeated for each $p$CO$_2$ treatment at both temperatures. This same experimental design was used for all species. This figure is taken from Bove et al. 2019.
(TIFF)

**S3 Fig. Experimental seawater parameters.** Calculated and measured seawater parameters over the entire experimental period.
(TIFF)

**S4 Fig. Reef and treatment PCAs by species.** Principal component analysis (PCA) of all coral physiological parameters for *S. siderea*, *P. strigosa*, and *P. astreoides* depicted by natal reef environment (**A-C**; offshore green, inshore yellow) and the combination of $p$CO$_2$ and temperature treatment (**D-F**). Arrows represent significant ($p < 0.05$) correlation vectors for physiological parameters and ellipses represent 95% confidence based on multivariate t-distributions.
(TIFF)

**S5 Fig. Measured physiological parameters per species.** Mean (±SE) physiological parameter (each row) measured for (**A**) *S. siderea*, (**B**) *P. strigosa*, and (**C**) *P. astreoides* at the completion of the 93-day experimental period. $p$CO$_2$ treatment is represented along the x axis and the temperature is depicted by colour (28˚C blue; 31˚C red).
(TIFF)

**S6 Fig. PCAs by reef and treatment across all species.** Principal component analysis (PCA) comparing the physiology of all three species at the end of the experiment depicted by (**A**) reef environment and (**B**) combined $p$CO$_2$ and temperature treatment. Arrows represent significant ($p < 0.05$) correlation vectors for physiological parameters and ellipses represent 95%

confidence based on multivariate t-distributions.
(TIFF)

**S7 Fig. Coral images through time per species.** Coral colour changes over the experimental period. Representative images of fragments of (**A**) *P. astreoides*, (**B**) *S. siderea*, and (**C**) *P. strigosa* from the same colonies demonstrating change in coral colour over time in either control (420 μatm; 28˚C) or warming (420 μatm; 31˚C) treatments from the start of the experiment (T0) to the end (T90).
(TIFF)

**S8 Fig. PCAs of *S. siderea* host or symbiont physiology.** Principal component analysis (PCA) of *S. siderea* coral host (protein, lipid, carbohydrate; left) or algal symbiont (chlorophyll a, symbiont density, colour intensity; right) physiological parameters by temperature (28˚C blue; 31˚C red), $p\text{CO}_2$ (pre industrial [300 μatm], light purple; current day [420 μatm], dark purple; end-of-century [680 μatm], light orange; extreme [3290 μatm], dark orange), and natal reef environment (offshore green; inshore yellow). Arrows represent significant ($p < 0.05$) correlation vectors for physiological parameters and ellipses represent 95% confidence based on multivariate t-distributions.
(TIFF)

**S9 Fig. PCAs of *P. strigosa* host or symbiont physiology.** Principal component analysis (PCA) of *P. strigosa* coral host (protein, lipid, carbohydrate; left) or algal symbiont (chlorophyll a, symbiont density, colour intensity; right) physiological parameters by temperature (28˚C blue; 31˚C red), $p\text{CO}_2$ (pre industrial [300 μatm], light purple; current day [420 μatm], dark purple; end-of-century [680 μatm], light orange; extreme [3290 μatm], dark orange), and natal reef environment (offshore green; inshore yellow). Arrows represent significant ($p < 0.05$) correlation vectors for physiological parameters and ellipses represent 95% confidence based on multivariate t-distributions.
(TIFF)

**S10 Fig. PCAs of *P. astreoides* host or symbiont physiology.** Principal component analysis (PCA) of *P. asteroides* coral host (protein, lipid, carbohydrate; left) or algal symbiont (chlorophyll a, symbiont density, colour intensity; right) physiological parameters by temperature (28˚C blue; 31˚C red), $p\text{CO}_2$ (pre industrial [300 μatm], light purple; current day [420 μatm], dark purple; end-of-century [680 μatm], light orange; extreme [3290 μatm], dark orange), and natal reef environment (offshore green; inshore yellow). Arrows represent significant ($p < 0.05$) correlation vectors for physiological parameters and ellipses represent 95% confidence based on multivariate t-distributions.
(TIFF)

**S1 Text. Document containing supplemental tables A through G with captions referenced in the main text.**
(PDF)

## Acknowledgments

We thank the Belize Fisheries Department for all associated permits, the Toledo Institute for Development and Environment (TIDE) and the Southern Environmental Association (SEA) for their support. We also thank S. Patel, S. Swinea, F. Buckthal, J. Townsend, J. Boulton, and C. Lopazanski for assisting with preparing corals for physiological assays and the Marchetti, Septer, and Waters labs at UNC Chapel Hill for equipment and lab space use.

## Author Contributions

**Conceptualization:** Colleen B. Bove, Sarah W. Davies, Justin B. Ries, Karl D. Castillo.

**Data curation:** Colleen B. Bove, Bailey C. Thomasson, Elizabeth B. Farquhar, Jess A. McCoppin.

**Formal analysis:** Colleen B. Bove, James Umbanhowar.

**Funding acquisition:** Colleen B. Bove, Justin B. Ries, Karl D. Castillo.

**Methodology:** Colleen B. Bove, Sarah W. Davies, Justin B. Ries, Karl D. Castillo.

**Supervision:** Sarah W. Davies, Justin B. Ries, Karl D. Castillo.

**Visualization:** Colleen B. Bove.

**Writing – original draft:** Colleen B. Bove.

**Writing – review & editing:** Colleen B. Bove, Sarah W. Davies, Justin B. Ries, James Umbanhowar, Bailey C. Thomasson, Elizabeth B. Farquhar, Jess A. McCoppin, Karl D. Castillo.

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
