## [Decision Letter · Decision Letter 0]

17 May 2022

PONE-D-22-07335Global change differentially modulates Caribbean coral physiology and suggests future ‘winners’ and ‘losers’PLOS ONE

Dear Dr. Bove,

Thank you for submitting your manuscript to PLOS ONE. After careful consideration, we feel that it has merit but does not fully meet PLOS ONE’s publication criteria as it currently stands. Therefore, we invite you to submit a revised version of the manuscript that addresses the points raised during the review process.

ACADEMIC EDITOR:Hello, 

   I have now had this article reviewed by two experts in the field. Although one was more favorable, the other necessitated a "major revision." I think most of the comments can be adequately addressed and so I am optimistic that this work can ultimately be published in PLoS ONE upon accommodating them. Looking forward to seeing the revised version in the coming weeks. Anderson

We look forward to receiving your revised manuscript.

Kind regards,

Anderson B. Mayfield, Ph.D.

Academic Editor

PLOS ONE

Journal Requirements:

“This research was partially supported by the Women Diver Hall of Fame Sea of Change Foundation Marine Conservation Scholarship and Lerner-Gray Memorial Fund of the American Museum of Natural History Grants for Marine Research awarded to CBB. JBR acknowledges support from NSF BIO-OCE award #1437371.”

“This research was partially supported by the Women Diver Hall of Fame Sea of Change Foundation Marine Conservation Scholarship (https://www.wdhof.org/scholarship/marine-conservation-scholarship-graduate) and Lerner-Gray Memorial Fund of the American Museum of Natural History Grants for Marine Research (https://www.amnh.org/research/richard-gilder-graduate-school/academics-and-research/fellowship-and-grant-opportunities/research-grants-and-graduate-student-exchange-fellowships/the-lerner-gray-fund-for-marine-research) awarded to CBB. JBR acknowledges support from NSF BIO-OCE award #1437371 (https://www.nsf.gov/geo/oce/programs/biores.jsp). The funders had no role in study design, data collection and analysis, decision to publish, or preparation of the manuscript.”

Additional Editor Comments:

Hello,

I have now had this article reviewed by two experts in the field. Although one was more favorable, the other necessitated a "major revision." I think most of the comments can be adequately addressed and so I am optimistic that this work can ultimately be published in PLoS ONE upon accommodating them. Looking forward to seeing the revised version in the coming weeks. Anderson

Reviewers' comments:

Reviewer's Responses to Questions

**Comments to the Author**

1. Is the manuscript technically sound, and do the data support the conclusions?

Reviewer #1: Yes

Reviewer #2: Partly

2. Has the statistical analysis been performed appropriately and rigorously? 

Reviewer #1: Yes

Reviewer #2: Yes

3. Have the authors made all data underlying the findings in their manuscript fully available?

Reviewer #1: Yes

Reviewer #2: Yes

4. Is the manuscript presented in an intelligible fashion and written in standard English?

Reviewer #1: Yes

Reviewer #2: Yes

5. Review Comments to the Author

Reviewer #1: Specific comments

It is important to understand the physiological effects of warming and acidification on coral and symbiotic algae. By this way, to understand the varied responses of the different symbiotic partners is critical for predicting the future of tropical coral reefs. This manuscript assess the physiological responses of three species of Caribbean corals come from inshore and offshore environments, and then independent and combined test under four ocean acidification conditions and two warming temperatures, with a long time experiments. These results help us understand the physiological responses and relationship between some species of corals and symbiotic algae under warming and acidification. And it require more investigation in multi-stressor, multi-species studies to explore these complexities of coral responses under global change.

There are still some questions about this manuscript. Are the morphology of the three corals the same (such as massive, branching...)? And will different morphology influences the response of physiological effects of warming and acidification on coral and symbiotic algae? In this experiments, the energy of coral completely came from the symbiotic algae. Will the physiological effects on corals and symbiotic algae keep the same or change if the coral get energy by feeding (heterotroph) under warming and acidification in field?

General comments

Comments while reading

Methods

Page 6. The authors did not describe or cite any references about how to control the PCO2 in the four CO2 treatments. How to create the pre-industrial CO2 condition and maintain it?

Page 9, line 226-228. How to see “no interactions” from S2 Table?

line 237. It suggests to use the full name, not the abbreviation in first time to mention noun. It replaces “ all PCs calculated above as… “ as “all Principal compound analysis (PCs) calculated above as …”.

Results

Page 11, line 266-267, 275, 285-286. How to read no significant interactive effect between temperature and pCO2 from S2 Table? Are the Fig S3D (DIC), S3E (HCO3), and S3F (CO2 seawater) cite here correct?

Page 11-12, line 289-297. The Figure 1 had A to F figures, however, the description only for A to C, without D to F. The figure reveal the symbol “sum”, the figure caption describe as sum/red = colour intensity. It is recommended to use uniform notation.

Discussion

Page 16-17, line 426-427. How to see “no interactions” from S2, S3, S8 Table?

Supplemental Materials for manuscript

S5 Table. I may confuse the results of S. sidera analysis. Did S. sidereal results only in offshore analysis, without any inshore analysis?

Reviewer #2: This paper studies the physiological response of three Caribbean coral species to two individually and combined global stressors (warming and acidification). Multiple physiological parameters in the corals are assessed using multidimensional analysis. Based on these data, the response of the coral species as well as 2 locations (inshore and offshore) are compared. The study includes a comprehensive amount of data on coral physiology that are of great value to understand the physiological response of these coral species to global stressors.

The paper could benefit from addressing in a more clear way (1) the ecological relevance of the experimental treatments and whether they reflect values that are relevant to Caribbean reefs, and (2) what the observed changes in physiology mean for the future of these coral species. In general, my main concern with the study is that the multivariable approach obscures the specific effects of the treatments on these corals species (or the lack of effects) and makes it harder to understand the implication of the stressors on the individual corals species and on the reefs of the future.

Specific comments:

Abstract: It would be useful to immediately introduce what species are considered “winners” or “losers” and why. Since “winners” and “losers” are in the title, it is expected to report this as the main result.

The introduction includes a nice description of the physiological parameters studied and how they have been seen to be affected by environmental stressors. This is very useful.

Results of previous studies about the relative susceptibility of Caribbean coral species to the stressors would be useful to offer some context. What species are becoming less or more abundant in locations that have experienced intense warming or OA?

Also, the introduction could benefit from including some context for the meaning and implications of physiological “plasticity”. In this paper, higher plasticity is assumed to imply higher susceptibility to environmental conditions. This should be better supported by adding some references to understand the results.

Methods: Generally, well organized. The authors provide enough details to understand the study and statistical analysis.

Lines 115 and 116 can be mentioned at the end of the paragraph for people that want more details, but perhaps these are not the best lines to start describing the study.

Lines 135-140 could become a table with the factorial CO2 and temperature conditions. From lines 135 to 138 it is not clear why 2 different values are presented for every CO2 treatment.

Line 200 mentioned that the images were analyzed for each timepoint, but I do not think there is a previous mention of when the time points are.

Lines 215-219. If mortality was 90-72% how many samples per coral species per treatment were actually analyzed? These numbers are more important than the 288 total samples reported in lines 130, if most of these 288 corals died before the physiological data was collected (tissue samples).

Results

The PCA section is clear and well organized. However, better organization of figure 1 is possible. For example, if different shapes are used for the CO2 treatments, colors can be used for the temperature (or vice versa) and then the top and bottom panels do not have to be duplicated. Since this is a fully factorial study it is perhaps better to visualize both treatment levels in one same panel.

In general, there are too many mentions to the supplementary information (Tables and Figures) in the results and discussion section. Is this necessary? It gives the feeling that half of the paper is actually not in the paper.

Lines 305-306. It is expected to have a strong correlation between symbiont density and Chl-a. Even more, I think you can argue that this variables are not independent and therefore you should include only one in your model. A more interesting and less redundant variable could be Chl-a content per Symbiont cell, and drop total Chl-a.

Discussion

Please elaborate some about how the studied treatments reflect the present or future conditions of these Caribbean corals.

Lines 449-450: Please provide some citations and explain how plasticity could be a detrimental response to stressors. Plasticity (Figure 3) is a big part of the discussion, but the negative connotation given to more plastic responses should be explained. Similarly, lines 455-457 could use references that support reduced capacity of corals with higher plasticity to persist under climate change scenarios.

Lines 490 and 524: photosynthetic efficiency was not presented as part of the data. Was it measured?

Small comments:

Writing style: “clearly” is used multiple times across the manuscript. This can be removed from most of the sentences and let the reader decide if something is clear based on the data. e.g., Line 395 could be: “pCO2 treatment drove differences in coral physiology” instead of “pCO2 treatment clearly drove differences in coral physiology”.

Lines 479-481 are repeated in the manuscript.

Thank you for making your data and code available.

6. PLOS authors have the option to publish the peer review history of their article (what does this mean?). If published, this will include your full peer review and any attached files.

Reviewer #1: No

Reviewer #2: No

---

## [Author Response · Author response to Decision Letter 0]

9 Jun 2022

Journal Requirements:

Response: We have updated our manuscript to meet PLOS ONE’s style requirements as requested.

“This research was partially supported by the Women Diver Hall of Fame Sea of Change Foundation Marine Conservation Scholarship and Lerner-Gray Memorial Fund of the American Museum of Natural History Grants for Marine Research awarded to CBB. JBR acknowledges support from NSF BIO-OCE award #1437371.”

“This research was partially supported by the Women Diver Hall of Fame Sea of Change Foundation Marine Conservation Scholarship (https://www.wdhof.org/scholarship/marine-conservation-scholarship-graduate) and Lerner-Gray Memorial Fund of the American Museum of Natural History Grants for Marine Research (https://www.amnh.org/research/richard-gilder-graduate-school/academics-and-research/fellowship-and-grant-opportunities/research-grants-and-graduate-student-exchange-fellowships/the-lerner-gray-fund-for-marine-research) awarded to CBB. JBR acknowledges support from NSF BIO-OCE award #1437371 (https://www.nsf.gov/geo/oce/programs/biores.jsp). The funders had no role in study design, data collection and analysis, decision to publish, or preparation of the manuscript.”

Response: We have removed the funding information from the text of the manuscript and the current funding statement should suffice. 

Response: We have added the following ethics statement to the methods section: “All corals were collected following local laws and regulations with appropriate permits (#5674).”

Response: We have added these captions for all supporting information at the end of the main text as requested and updated the in-text citations.

Review Comments to the Author

Reviewer #1:

Specific comments

It is important to understand the physiological effects of warming and acidification on coral and symbiotic algae. By this way, to understand the varied responses of the different symbiotic partners is critical for predicting the future of tropical coral reefs. This manuscript assess the physiological responses of three species of Caribbean corals come from inshore and offshore environments, and then independent and combined test under four ocean acidification conditions and two warming temperatures, with a long time experiments. These results help us understand the physiological responses and relationship between some species of corals and symbiotic algae under warming and acidification. And it require more investigation in multi-stressor, multi-species studies to explore these complexities of coral responses under global change.

There are still some questions about this manuscript. Are the morphology of the three corals the same (such as massive, branching...)? And will different morphology influences the response of physiological effects of warming and acidification on coral and symbiotic algae? In this experiments, the energy of coral completely came from the symbiotic algae. Will the physiological effects on corals and symbiotic algae keep the same or change if the coral get energy by feeding (heterotroph) under warming and acidification in field?

Response: We thank the reviewer for taking time to review our manuscript and provide helpful feedback to improve it. We have addressed your comments and provided clarity on some of your concerns throughout the manuscript. The morphology of all three coral species can be classified as mounding so we do not anticipate differences in morphology to have an impact on the responses of these species. This is important information and therefore we clarified that these are all mounding coral species in the introduction.

Lines 118-121 now state: “These coral species were selected because they represent both weedy (P. astreoides) and stress-tolerant (S. siderea and P. strigosa) life histories [38], possess similar growth morphologies (mounding), and are common throughout the Caribbean.”

In regards to your comment on feeding of the corals in this experiment, the corals were fed throughout the experiment to support heterotrophy, however, this was not stated in the methods text and we appreciate the reviewer pointing this out. We have now added the following clarifying statement into the Experimental design section of the Methods: 

Lines 169-170 now state: “Corals were fed a combination of frozen adult Artemia sp. and newly hatched Artemia sp. every other day to satisfy heterotrophic feeding.”

Methods

Page 6. The authors did not describe or cite any references about how to control the PCO2 in the four CO2 treatments. How to create the pre-industrial CO2 condition and maintain it?

Response: We have now added the following statement into the methods to describe how we achieved pCO2 treatments. 

Lines 158-161 now state: “High-precision digital solenoid-valve mass flow controllers (Aalborg Instruments and Controls; Orangeburg, NY, USA) were used to bubble air alone (control pCO2 conditions), or in combination with CO2-free air (pre-industrial conditions) or CO2 gas (end-of-century and extreme conditions) to achieve gas mixtures of each desired pCO2 condition.”

Page 9, line 226-228. How to see “no interactions” from S2 Table?

Response: We agree this was unclear so we have clarified this statement. 

Lines 252-254 now state: "The additive model resulted in a lower AIC than the fully interactive model for all species, so interaction terms were dropped from each model resulting in fully additive models (see S2 Table)."

line 237. It suggests to use the full name, not the abbreviation in first time to mention noun. It replaces “ all PCs calculated above as… “ as “all Principal compound analysis (PCs) calculated above as …”.

Response: We have updated this to use the full name (principal components) instead of just the abbreviation here.

Results

Page 11, line 266-267, 275, 285-286. How to read no significant interactive effect between temperature and pCO2 from S2 Table? Are the Fig S3D (DIC), S3E (HCO3), and S3F (CO2 seawater) cite here correct?

Response: We have clarified this statement as recommended above. We did incorrectly reference S3D Fig in the supplements, when we meant to reference S4D and we thank the reviewer for noticing this. 

Page 11-12, line 289-297. The Figure 1 had A to F figures, however, the description only for A to C, without D to F. The figure reveal the symbol “sum”, the figure caption describe as sum/red = colour intensity. It is recommended to use uniform notation.

Response: We thank the reviewer for highlighting the mistake in our figure caption, which was missing several labels and we have now fixed the caption text. In addition, we have fixed the colour label as requested so now all PC loadings use 'colour'

Discussion

Page 16-17, line 426-427. How to see “no interactions” from S2, S3, S8 Table?

Response: We reference these tables because these represent our model selection process and highlight that we do not identify significant interaction terms in any of our models. However, at the request of Reviewer 2, we have removed this reference to reduce the number of references to the supplemental materials.

Supplemental Materials for manuscript

S5 Table. I may confuse the results of S. sidera analysis. Did S. sidereal results only in offshore analysis, without any inshore analysis?

Response: The analysis of S. siderea was the only one to include reef environment in the model, but both reef environments were included in the analysis. The intercept coefficient for S. siderea represents inshore fragments for comparison across other coefficients. This is highlighted in the figure caption to assist the reader.

Reviewer #2: 

This paper studies the physiological response of three Caribbean coral species to two individually and combined global stressors (warming and acidification). Multiple physiological parameters in the corals are assessed using multidimensional analysis. Based on these data, the response of the coral species as well as 2 locations (inshore and offshore) are compared. The study includes a comprehensive amount of data on coral physiology that are of great value to understand the physiological response of these coral species to global stressors.

The paper could benefit from addressing in a more clear way (1) the ecological relevance of the experimental treatments and whether they reflect values that are relevant to Caribbean reefs, and (2) what the observed changes in physiology mean for the future of these coral species. In general, my main concern with the study is that the multivariable approach obscures the specific effects of the treatments on these corals species (or the lack of effects) and makes it harder to understand the implication of the stressors on the individual corals species and on the reefs of the future.

Response: We thank the reviewer for taking their time to review this manuscript and provide very helpful feedback that has improved our manuscript. We especially thank the reviewer for also highlighting things they appreciated about our manuscript in addition to highlighting areas for improvement.

Specific comments:

Abstract: It would be useful to immediately introduce what species are considered “winners” or “losers” and why. Since “winners” and “losers” are in the title, it is expected to report this as the main result.

Response: We have now added the following statement into the abstract.

Lines 43-46 now state: “Further, our study identifies S. siderea and P. astreoides as potential ‘winners’ on future Caribbean coral reefs due to their resilience under projected global change stressors, while P. strigosa will likely be a ‘loser’ due to their sensitivity to thermal stress events.”

The introduction includes a nice description of the physiological parameters studied and how they have been seen to be affected by environmental stressors. This is very useful.

Response: Thank you!

Results of previous studies about the relative susceptibility of Caribbean coral species to the stressors would be useful to offer some context. What species are becoming less or more abundant in locations that have experienced intense warming or OA?

Response: We have added several statements about Caribbean species under these stressors into the intro:

Lines 63-67 now state: “For example, the Caribbean coral species Siderastrea siderea and Porites astreoides have been shown previously to maintain higher growth rates under ocean acidification and/or warming stress, [16–18] and other species, such as Orbicella faveolata and Acropora cervicornis, generally exhibited reduced growth under these same stressors [14,18,19].”

Lines 118-122 now state: “These coral species were selected because they represent both weedy (P. astreoides) and stress-tolerant (S. siderea and P. strigosa) life histories [45], possess similar growth morphologies (mounding), and are common throughout the Caribbean. These coral species are common throughout the Caribbean and can be found across a variety of environmental gradients.”

Also, the introduction could benefit from including some context for the meaning and implications of physiological “plasticity”. In this paper, higher plasticity is assumed to imply higher susceptibility to environmental conditions. This should be better supported by adding some references to understand the results.

Response: We have now added the following paragraph into the introduction to provide a bit more context about physiological plasticity

Lines 101-112 now state: “Many symbiotic corals also have the capacity to exhibit physiological plasticity (i.e., modification of an organism’s physiology) in response to changing environments that may be employed under global change scenarios [38,39]. While plasticity is often highlighted as a mechanism for rapid response to changing environments, there is still debate about whether plasticity alone is enough to ensure species persistence under global change [40]. Indeed, a highly plastic coral may be able to modulate its physiology (e.g., increase chlorophyll a per symbiont cell) under an acute stress event (e.g., low light levels) [41], but this is likely to come at a cost to another metabolic process, such as energy stores. This physiological cost can be beneficial for the coral in the short term, however, may eventually result in a decline in fitness [40,42], especially in long-lived organisms like reef-building corals. These potential trade-offs in reef-building corals remain poorly understood and highlight the complexities of plasticity as a mode of global change resilience.”

Methods

Generally, well organized. The authors provide enough details to understand the study and statistical analysis.

Response: Thank you!

Lines 115 and 116 can be mentioned at the end of the paragraph for people that want more details, but perhaps these are not the best lines to start describing the study.

Response: We have moved this sentence to the end of the paragraph as requested.

Lines 135-140 could become a table with the factorial CO2 and temperature conditions. From lines 135 to 138 it is not clear why 2 different values are presented for every CO2 treatment.

Response: We have now added the below table to the main text to describe the experimental treatments (Table 1). Regarding the two different values, because temperature affects the solubility of CO2 in seawater, the two temperature treatments averaged different carbonate parameters for each of the pCO2 treatments, despite being sparged with the same gas mixture ratios.

Table 1. Warming and acidification treatment means and standard deviations. 

Temperature Treatment (°C) pCO2 treatment (µatm)

 Pre-industrial Current-day End-of-century Extreme

Control 28 ± 0.4 288 ± 65 47 ± 152 673 ± 104 3285 ± 484

Warming 31 ± 0.4 31 ± 96 405 ± 91 701 ± 94 3309 ± 414

Line 200 mentioned that the images were analyzed for each timepoint, but I do not think there is a previous mention of when the time points are.

Response: Thank you for catching this, we have clarified this statement in the text

Lines 225-226 now state: “Coral color intensity was also analysed from images of every fragment with standardized color scales taken every 30 days throughout the experiment.”

Lines 215-219. If mortality was 90-72% how many samples per coral species per treatment were actually analyzed? These numbers are more important than the 288 total samples reported in lines 130, if most of these 288 corals died before the physiological data was collected (tissue samples).

Response: We have updated this statement to report the total number of analyzed fragments per species.

Lines 241-245 now state: “Sample mortality was observed throughout the experimental period across species as described in Bove et al. [17] and thus some treatments resulted in reduced replication for physiological analyses (Table A in S1 Text). Overall, S. siderea exhibited nearly 90% survival (86 total fragments), P. strigosa exhibited 80% survival (77 total fragments), and P. astreoides exhibited 72% survival (69 total fragments) at the end of the experiment [17].”

Results

The PCA section is clear and well organized. However, better organization of figure 1 is possible. For example, if different shapes are used for the CO2 treatments, colors can be used for the temperature (or vice versa) and then the top and bottom panels do not have to be duplicated. Since this is a fully factorial study it is perhaps better to visualize both treatment levels in one same panel.

Response: We appreciate this recommendation for this figure and have condensed the 6-panel figure into 3 (one PCA per species). Since S. siderea and P. astreoides were mostly explained by pCO2 treatment, these data are coloured by pCO2 treatment, while P. strigosa was driven by temperature treatment so we used colour to represent this pattern. 

In general, there are too many mentions to the supplementary information (Tables and Figures) in the results and discussion section. Is this necessary? It gives the feeling that half of the paper is actually not in the paper.

Response: While we understand the reviewer’s comment about referencing the supplemental materials, we feel that the materials in the supplemental document are necessary. These tables and figures represent analyses and visualisations we explored to support our overall conclusions from our study. We want to be transparent about the different ways we explored our data but also maintain the manuscript figures succinct and easy to digest. We have removed some duplicate references to the supplemental materials in an effort to reduce the feeling that most of the paper is missing.

Lines 305-306. It is expected to have a strong correlation between symbiont density and Chl-a. Even more, I think you can argue that this variables are not independent and therefore you should include only one in your model. A more interesting and less redundant variable could be Chl-a content per Symbiont cell, and drop total Chl-a.

Response: We chose to include both symbiont density and chlorophyll a in these correlation analyses to compare the correlations across all parameters to see how they change between species. For example, we report a strong correlation between these parameters in S. siderastrea (R2 = 0.72), while these parameters are not highly correlated in P. astreoides (R2 = 0.20). Further, we considered the chla/cell assessment, however, this metric produces concerning values when measuring low symbiont counts (i.e., in bleached P. strigosa) where chlorophyll a was still present, but counts were very low (see below figure where this is depicted). For this reason, we decided to measure these parameters individually and discuss the implications of high density or chlorophyll content for the coral. 

Discussion

Please elaborate some about how the studied treatments reflect the present or future conditions of these Caribbean corals.

Response: We have provided these details within the methods section.

Lines 153-158 now state: “The eight treatments encompassed four pCO2 treatments (Table 1) corresponding to pre-industrial, current-day (pCO2 control), moderate end-of-century, and an extreme pCO2 level all crossed with two temperatures (Table 1) corresponding to the corals’ approximate present-day summer mean (28°C ±0.4) and projected end-of-century summer warming (31°C ±0.4) [49] that has also been observed to induce bleaching in these species [50].”

Lines 449-450: Please provide some citations and explain how plasticity could be a detrimental response to stressors. Plasticity (Figure 3) is a big part of the discussion, but the negative connotation given to more plastic responses should be explained. Similarly, lines 455-457 could use references that support reduced capacity of corals with higher plasticity to persist under climate change scenarios.

Response: We have added additional references to support our statements as requested. Additionally, we have explained our rationale behind our conclusions here further.

Lines 476-480 now state: “However, physiological plasticity may not always be beneficial long term and may instead signal a shift in organism condition [18,42,44]. Organisms exhibiting higher plasticity in response to environmental change (e.g., ocean warming and acidification) may incur a physiological cost in the form of a trade-off that ultimately may impact the population’s ability to resist future change [40–42].”

Lines 490 and 524: photosynthetic efficiency was not presented as part of the data. Was it measured?

Response: This was not assessed here and we have clarified the statement in the text since this was in reference to chlorophyll a content.

Lines 517-520 now state: “This pattern suggests P. strigosa are consuming carbohydrate and protein stores in response to reduced symbiont density and chlorophyll a content, while lipid stores remain relatively unaltered, in line with previous work on coral energetics”

Small comments:

Writing style: “clearly” is used multiple times across the manuscript. This can be removed from most of the sentences and let the reader decide if something is clear based on the data. e.g., Line 395 could be: “pCO2 treatment drove differences in coral physiology” instead of “pCO2 treatment clearly drove differences in coral physiology”.

Response: We have removed many uses of ‘clearly’ from the text per request. 

Lines 479-481 are repeated in the manuscript.

Response: We have now modified these statements like so: 

Lines 465-467 now state: “It is clear that coral responses under global change remain complex and require further investigation using additional multi-stressor, multi-species studies to tease apart these complexities.”

Lines 508-510 now state: “Either way, the role of plasticity in coral responses to global change is complex and merits further investigation to better understand species-specific levels of resilience.”

Thank you for making your data and code available.

Response: Thank you for the kind comment here, it is a major goal of ours to ensure open and transparent science.

---

## [Decision Letter · Decision Letter 1]

13 Jul 2022

PONE-D-22-07335R1Global change differentially modulates Caribbean coral physiology and suggests future ‘winners’ and ‘losers’PLOS ONE

Dear Dr. Bove,

Thank you for submitting your manuscript to PLOS ONE. After careful consideration, we feel that it has merit but does not fully meet PLOS ONE’s publication criteria as it currently stands. Therefore, we invite you to submit a revised version of the manuscript that addresses the points raised during the review process.

We look forward to receiving your revised manuscript.

Kind regards,

Anderson B. Mayfield, Ph.D.

Academic Editor

PLOS ONE

Journal Requirements:

Additional Editor Comments :

Hello,

I apologize for this taking so long, but one of the original reviewers was too busy to review it, and then 13 others declined (likely because they were in Bremen)! But I digress. The new reviewer has raised some minor issues, but I don't see them as being an issue. I'll be looking forward to seeing the revised version in a few weeks.

Thanks,

Anderson

Reviewers' comments:

Reviewer's Responses to Questions

**Comments to the Author**

1. If the authors have adequately addressed your comments raised in a previous round of review and you feel that this manuscript is now acceptable for publication, you may indicate that here to bypass the “Comments to the Author” section, enter your conflict of interest statement in the “Confidential to Editor” section, and submit your "Accept" recommendation.

Reviewer #1: (No Response)

Reviewer #3: (No Response)

2. Is the manuscript technically sound, and do the data support the conclusions?

Reviewer #1: (No Response)

Reviewer #3: Yes

3. Has the statistical analysis been performed appropriately and rigorously? 

Reviewer #1: (No Response)

Reviewer #3: Yes

4. Have the authors made all data underlying the findings in their manuscript fully available?

Reviewer #1: (No Response)

Reviewer #3: Yes

5. Is the manuscript presented in an intelligible fashion and written in standard English?

Reviewer #1: (No Response)

Reviewer #3: Yes

6. Review Comments to the Author

Reviewer #1: (No Response)

Reviewer #3: Please see the uploaded word document for general comments as well as specific comments to the authors.

7. PLOS authors have the option to publish the peer review history of their article (what does this mean?). If published, this will include your full peer review and any attached files.

Reviewer #1: No

Reviewer #3: No

---

## [Author Response · Author response to Decision Letter 1]

8 Aug 2022

Review Comments to the Author

General comments:

This paper is a revised version of an initial submission, though this reviewer was not one of the two original reviewers. Overall, I think this paper is improved from its original version based on the previous reviewers’ comments and the tracked changes. I do think this paper deserves to be published in PLoS ONE, but recommend some additional minor revisions prior to acceptance. 

I appreciate the Github link for the data and analysis codes used for this paper.

Response: Thank you for the helpful feedback and we appreciate the kind comment regarding the inclusion of the GitHub link to the data and code. We strive for transparency and reproducibility! 

Specific comments:

I think the introduction is well-written, especially after seeing the additions made from the first round of review, but I have a few additional comments.

Line 119: I would classify S.sid as “weedy” with P.ast

Response: We are using the classifications for these species based on the study led by Darling in 2012 where they assign these labels across coral species. We agree that S. siderea appears to exhibit weedy traits, but it is classified as a stress-tolerant species based on the referenced work.

Line 119: in reference to line 435 in the discussion – in the discussion the authors write “Indeed, P. strigosa is known to be a more thermally sensitive coral species…” but in line 119 in the intro you write that P.strig was chosen b/c it’s stress-tolerant. This needs to be clarified.

Response: We have left the classification of P. strigosa as ‘stress tolerant’ in the intro because this was our reasoning behind selecting this species. However, recent work has demonstrated the sensitivity of this species in more recent years. We have now updated this portion of the discussion to cover this shift (Lines 463-467): “Indeed, while P. strigosa was previously classified as a stress-tolerant species based on trait assessment [45], it has more recently been identified as a more thermally sensitive coral species [35,68,69]. This response is likely representative of the overall deterioration of coral condition in response to thermal stress, which may lead to mortality under chronic or extreme exposure as is being seen more frequently on Caribbean coral reefs [5].”

Line 119: Maybe the authors could consider expanding upon the framing of the 3 selected species – we know that coral cover has declined rapidly on Caribbean reefs due to a myriad of factors, most recent of which include SCTLD. Perhaps framing the experiment a little more intentionally, e.g., you selected weedy/stress tolerant spp because you wanted to test what the predominant responses of the likely-to-be-dominant corals on reefs of the Caribbean will be… it’s not far off from what you have now, but right now it feels like it’s implied – like you didn’t write that but I’m guessing that’s what you intended. If you’re more intentional with the rationale and the “why it matters”, it just makes the reader more invested in reading on. 

Response: We appreciate this suggestion to be more explicit here about the framing of the study and so we have updated this section of the introduction to read (Lines 120-127): “These coral species were selected because they represent both weedy (P. astreoides) and stress-tolerant (S. siderea and P. strigosa) life histories [45], possess similar growth morphologies (mounding), and are common throughout the Caribbean . These coral species are common throughout the Caribbean and can be found across a variety of environmental gradients. Additionally, we included corals from two distinct reef environments to assess how environmental histories impact responses to global change stressors. Overall, we selected these species to better understand how corals that are expected to dominate Caribbean reefs in the future may respond to global change stressors.”

Truthfully, when I first read this paper, my initial thought was this deserves to be published; however, I’m not sure how novel it is. Lots of labs have done cross-factorial TxCO2 work like this to assess physiological responses in corals and their symbionts between 2012-present (other Bove papers, other Castillo lab papers, Baumann, Grottoli, Schoepf, Towle, Okazaki, Manzello, Enochs, Edmunds, and many more…) But the framing of this is what might set this paper apart from the many that came before it. 

We know the Caribbean is in trouble. We know we have lost a lot of major reef builders. The question that persists is: for the weedy and stress tolerant spp – is there hope for them under combined and possibly extreme TXCO2 scenarios, and on top of that – at prolonged stress exposure, e.g. 90 days, not just 30 or 60 days? I’d like to see the authors play that up and distinguish how this work is novel compared to the many previous studies that at first glance might seem very similar. Convince us why this experiment matters – why we still need this research as a contribution to the literature. The authors start to get at this in lines 593-599 in the Discussion, but I think expanding it could really strengthen the paper.

Response: We appreciate your support of this work and understand that we need to highlight how this work is unique more throughout the manuscript. We have added several statements throughout to drive this message home to readers, for example:

Lines 438-440: “Caribbean coral reefs have experienced considerable shifts in ecosystem composition since the 1970s defined by declines in several stony coral taxa [67,68], resulting in reefs now dominated by weedy and stress-tolerant species.”

Lines 497-500: “Further, while other studies report synergistic effects on coral physiology, most of these studies only assess a single parameter, potentially missing other key physiological responses that suggest more additive responses like observed here.”

Lines 652-654: “As global change continues, it is critical to understand species-specific responses of coral to ocean acidification and warming scenarios to predict the future of Caribbean coral reef assemblages, especially with a focus on now-dominant coral species explored here. “

Lines 657-660: “Conversely, the previously assumed stress-tolerant species P. strigosa was unable to maintain any physiological traits under warming, suggesting that this species is now particularly vulnerable to thermal stress, which will likely lead to widespread bleaching and mortality.”

Methods:

Line 163, Table 1: Are there two typos?

In the preindustrial CO2 column should the 31 +/- 96 be 301?

Similarly in the current-day CO2 column should the 47 +/- 152 be 407?

Response: Thank you for catching these typos, we have fixed them according to the data (31 should have been 311 and 47 should have been 447).

Line 169-170: Is there a reason why the authors combined frozen artemia and newly hatched artemia? Why not just keep the feedings consistent by using one or the other? Doesn’t that technically introduce another variable? In reality, I’m sure there was no measurable effect on heterotrophy between frozen and freshly hatched, but you never know…? Also – how much artemia were fed to each tank? Quantity is important to mention here when thinking about how feeding affects lipids, etc., even though feeding rate was not tested in this study. In short, I noticed the sentence (line 169-170) was added following the first review, but I still think it could use an additional sentence or two of clarification about amount (quantity) of heterotrophic food offered to the corals in each treatment tank. Do the authors know approximate zooplankton density on the natal reefs in Belize? Or was the amount fed based on previous studies?

Response: We combined the frozen and newly hatched because we wanted to make sure that there were different size classes available to the corals and because there was some concern that freshly-hatched Artemia would not have the same nutritional value to more mature Artemia. The frozen and freshly hatched artemia were combined together into a container before being added to the experimental tanks every time so each system was provided the same mixture every time, making it consistent across treatments and tanks. We have clarified this and the amount fed in the text as requested, as well as included the references that helped us settle on this feeding regime and the section now reads (Lines 175-177): “Corals were fed a combination of ca. 6 g frozen adult Artemia and 250 mL concentrated newly hatched live Artemia (500 mL-1) every other day to satisfy heterotrophic feeding [51,52].”

Results:

I noticed a couple of inconsistencies with italicizing and using the subscript “2” for CO2 so I recommend the authors just do a find and replace for species names and “CO2”. Examples include line 316 (species names not italicized) and line 320 “pCO2”. I also noticed use of the British spelling for “colour”. I think as long as it’s consistent it’s fine, and I’m sure if the journal wants the use of “color” that can be corrected during final formatting/editing, but just flagging that the authors might want to check.

Response: Thank you for catching this inconsistency, we have changed the spelling throughout to use ‘colour’ unless the journal prefers the American English spelling. Additionally, we have fixed the use of CO2 throughout as well!

Line 369: Do you have any ideas about why the P. strigosa controls had low survivorship? Do you attribute it to transfer stress form Belize to Massachusetts? Having low survivorship in the controls is a reason to potentially be more critical (or at least skeptical) of treatment results, so if the authors have any ideas about this, they should be briefly mentioned/discussed.

Response: The starting N for this system was lower at the start of the experiment because of loss of samples through the adjustment period. We noticed this system had been impacted by some microorganisms so we had to construct a new ‘control’ system for all species, however, we had many fewer reserve fragments of these genotypes for this species so lower N to begin with. We have thus added the following sentence into the methods to address this (Lines 260-263): “Further, the initial and final control treatment sample size of P. strigosa was lower than other species because this treatment system had to be reconstructed before the start of the experiment and there were only a few reserve genotypes of this species available for the new control system.”

Line 401: Does “constrained physiology” mean “less plastic”?

Response: Yes, we interpret this constrained physiology to mean these corals are less plastic and that corals under these conditions will exhibit similar responses.

Discussion:

Line 435: see comment above from introduction about the incongruence with the statement in line 119.

Response: We have addressed this comment as requested (detailed above).

Line 494-510: I found this paragraph a little hard to get through. There’s so much use of the word “plasticity” and the positive and negative aspects of how plasticity manifested in P. ast and P. strig in the experiment. Then the authors distinguish between the concept of “physiological plasticity” from the study and genotypic plasticity (not analyzed in this study). I get what the authors are trying to convey, but the wording and flow of this paragraph (at least for me) could be improved to help the reader more quickly grasp the take-home messages.

Response: We have updated the text in this paragraph to improve readability and flow as requested. This section how reads (Lines 529-544): “Varying levels of plasticity in P. strigosa and P. astreoides from different habitats has been previously reported [47,86]; however, natal reef effects were not evident in either species in this study (Fig 3B-C). The small sample size of P. strigosa likely contributed to the lack of differences between habitats, while different measures of plasticity – physiological plasticity (present study) vs. gene expression plasticity [86] – may contribute to the inconsistent responses observed in P. astreoides. While neither species exhibited differing levels of plasticity between reef environments, both P. strigosa and P. astreoides appear to exhibit higher plasticity at the elevated temperature, though this is only statistically significant in P. astreoides (Fig 3B-C). Interestingly, the higher plasticity at elevated temperatures in P. strigosa was associated with diminished physiological conditions, while higher plasticity in P. astreoides manifested as improved physiology (Fig 1B-C). These differences highlight how plasticity may result from physiological trade-offs in response to environmental change in some organisms (i.e., P. strigosa) [42,85], while other organisms (i.e., P. astreoides) may benefit from such plastic responses to match their physiology to their environment [87]. Either way, the role of plasticity in coral responses to global change is complex and merits further investigation to better understand species-specific levels of resilience.”

Line 590-593: I generally agree with this statement. I realize you tested more than one parameter in this study, but you still proceeded to use the “winner” and “loser” terminology in your article title after cautioning others against this (for good reason)…

Response: We agree with your concern about this statement and continuing to use those terms in the title so we have updated the title to be “Global change differentially modulates Caribbean coral physiology”.

Lines 593-599: See earlier comment about expansion of these ideas.

Response: We had expanded the discussion as requested and highlighted some of these areas in our response above.

---

## [Decision Letter · Decision Letter 2]

18 Aug 2022

Global change differentially modulates Caribbean coral physiology

PONE-D-22-07335R2

Dear Dr. Bove,

We’re pleased to inform you that your manuscript has been judged scientifically suitable for publication and will be formally accepted for publication once it meets all outstanding technical requirements.

Kind regards,

Anderson B. Mayfield, Ph.D.

Academic Editor

PLOS ONE

Additional Editor Comments (optional):

Reviewers' comments:

Reviewer's Responses to Questions

**Comments to the Author**

1. If the authors have adequately addressed your comments raised in a previous round of review and you feel that this manuscript is now acceptable for publication, you may indicate that here to bypass the “Comments to the Author” section, enter your conflict of interest statement in the “Confidential to Editor” section, and submit your "Accept" recommendation.

Reviewer #1: All comments have been addressed

Reviewer #3: All comments have been addressed

2. Is the manuscript technically sound, and do the data support the conclusions?

Reviewer #1: Yes

Reviewer #3: Yes

3. Has the statistical analysis been performed appropriately and rigorously? 

Reviewer #1: Yes

Reviewer #3: Yes

4. Have the authors made all data underlying the findings in their manuscript fully available?

Reviewer #1: Yes

Reviewer #3: Yes

5. Is the manuscript presented in an intelligible fashion and written in standard English?

Reviewer #1: Yes

Reviewer #3: Yes

6. Review Comments to the Author

Reviewer #1: (No Response)

Reviewer #3: Thanks for addressing all of the comments in the re- review. I think the authors have done so adequately and appropriately. I think the title change is good, and appreciate the clarifications on the heterotrophic feedings as well as the efforts to distinguish this work from previous, similar studies. I believe this is now suitable for publication in PLoS ONE.

7. PLOS authors have the option to publish the peer review history of their article (what does this mean?). If published, this will include your full peer review and any attached files.

Reviewer #1: No

Reviewer #3: No

---

## [Editor Report · Acceptance letter]

24 Aug 2022

PONE-D-22-07335R2 

Global change differentially modulates Caribbean coral physiology 

Dear Dr. Bove:

I'm pleased to inform you that your manuscript has been deemed suitable for publication in PLOS ONE. Congratulations! Your manuscript is now with our production department. 

Kind regards, 

on behalf of

Dr. Anderson B. Mayfield 

Academic Editor

PLOS ONE